# Consistent Jumpy Predictions for Videos and Scenes

## Abstract

Stochastic video prediction models take in a sequence of image frames, and generate a sequence of consecutive future image frames. These models typically generate future frames in an autoregressive fashion, which is slow and requires the input and output frames to be consecutive. We introduce a model that overcomes these drawbacks by generating a latent representation from an arbitrary set of frames that can then be used to simultaneously and efficiently sample temporally consistent frames at arbitrary time-points. For example, our model can "jump" and directly sample frames at the end of the video, without sampling intermediate frames. Synthetic video evaluations confirm substantial gains in speed and functionality without loss in fidelity. We also apply our framework to a 3D scene reconstruction dataset. Here, our model is conditioned on camera location and can sample consistent sets of images for what an occluded region of a 3D scene might look like, even if there are multiple possibilities for what that region might contain. Reconstructions and videos are available at `https://bit.ly/2O4Pc4R`.

## 1 Introduction

The ability to fill in the gaps in high-dimensional data is a fundamental cognitive skill. Suppose you glance out of the window and see a person in uniform approaching your gate carrying a letter. You can easily imagine what will (probably) happen a few seconds later. The person will walk up the path and push the letter through your door. Now suppose you glance out of the window the following day and see a person in the same uniform walking down the path, away from the house. You can easily imagine what (probably) happened a few seconds earlier. The person came through your gate, walked up the path, and delivered a letter. Moreover, in both instances, you can visualize the scene from different viewpoints. From your vantage point at the window, you can imagine how things might look from the gate, or from the front door, or even from your neighbour's roof. Essentially, you have learned from your past experience of unfolding visual scenes how to flexibly extrapolate and interpolate in both time and space. Replicating this ability is a significant challenge for AI.

To make this more precise, let's first consider the traditional video prediction setup. Video prediction is typically framed as a sequential forward prediction task. Given a sequence of frames $f_1, ..., f_t$, a model is tasked to generate future frames that follow, $f_{t+1}, ..., f_T$. Existing models (Babaeizadeh et al., 2018; Kalchbrenner et al., 2017; Finn et al., 2016) carry out the prediction in an autoregressive manner, by sequentially sampling frame $f_{t+n}$ from $p(f_{t+n}|f_1, ..., f_{t+n-1})$.

This traditional setup is often limiting – in many cases, we are interested in what happens a few seconds after the input sequence. For example, in model based planning tasks, we might want to predict what frame $f_T$ is, but might not care what intermediate frames $f_{t+1}, ..., f_{T-1}$ are. Existing models must still produce the intermediate frames, which is inefficient. Instead, our model is "jumpy" – it can directly sample frames in the future, bypassing intermediate frames. For example, in a 40-frame video, our model can sample the final frame 12 times faster than an autoregressive model like SV2P (Babaeizadeh et al., 2018).

More generally, existing forward video prediction models are not flexible at filling in gaps in data. Instead, our model can sample frames at arbitrary time points of a video, given a set of frames at arbitrary time points as context. So it can, for example, be used to infer backwards or interpolate between frames, as easily as forwards. Our model is also not autoregressive on input or output frames – it takes in each input frame in parallel, and samples each output frame in parallel.

In our setup of "jumpy" video prediction, a model is given frames $f_1, ..., f_n$ from a single video along with the arbitrary time-points $t_1, ..., t_n$ at which those frames occurred. To be clear, $t_1, ..., t_n$ need not be consecutive or even in increasing order. The model is then asked to sample plausible frames at arbitrary time points $t'_1, ..., t'_k$. In many cases there are multiple possible predicted frames given the context, making the problem stochastic. For example, a car moving towards an intersection could turn left or turn right. Although our model is not autoregressive, each set of $k$ sampled frames is consistent with a single coherent possibility, while maintaining diversity across sets of samples. That is, in each sampled set all $k$ sampled frames correspond to the car moving left, or all correspond to the car moving right.

At a high level, our model, JUMP, takes in the input frames and samples a stochastic latent that models the stochasticity in the video. Given an arbitrary query time-point, the model uses the sampled latent to render the frame at that time-point. The model is not autoregressive on input or output frames, but still captures correlations over multiple target frames. Our method is not restricted to video prediction. When conditioned on camera position, our model can sample consistent sets of images for an occluded region of a scene, even if there are multiple possibilities for what that region might contain.

We test JUMP on multiple synthetic datasets. To summarize our results: on a synthetic video prediction task involving five shapes, we show that JUMP produces frames of a similar image quality to modern video prediction methods. Moreover, JUMP converges more reliably, and can do jumpy predictions. To showcase the flexibility of our model we also apply it to stochastic 3D reconstruction, where our model is conditioned on camera location instead of time. For this, we introduce a dataset that consists of images of 3D scenes containing a cube with random MNIST digits engraved on each of its faces. JUMP outperforms GQN (Eslami et al., 2018) on this dataset, as GQN is unable to capture several correlated frames of occluded regions of the scene. We focus on synthetic datasets so that we can quantify how each model deals with stochasticity. For example, in our 3D reconstruction dataset we can control when models are asked to sample image frames for occluded regions of the scene, to quantify the consistency of these samples. We strongly encourage the reader to check the project website `https://bit.ly/2O4Pc4R` to view videos of our experiments.

To summarize, our key contributions are as follows.

1. *We motivate and formulate the problem of jumpy stochastic video prediction*, where a model has to predict consistent target frames at arbitrary time points, given context frames at arbitrary time points. We observe close connections with consistent stochastic 3D reconstruction tasks, and abstract both problems in a common framework.

2. *We present a model for consistent jumpy predictions in videos and scenes*. Unlike existing video prediction models, our model consumes input frames and samples output frames entirely in parallel. It enforces consistency of the sampled frames by training on multiple correlated targets, sampling a global latent, and using a deterministic rendering network.

3. *We show strong experimental results for our model*. Unlike existing sequential video prediction models, our model can also do jumpy video predictions. We show that our model also produces images of similar quality while converging more reliably. Our model is not limited to video prediction – we develop a dataset for stochastic 3D reconstruction. Here, we show that our model significantly outperforms GQN.

## 2 RELATED WORK

**Video Prediction** A common limitation of video prediction models is their need for determinism in the underlying environments (Lotter et al., 2017; Srivastava et al., 2015; Boots et al., 2014; Finn et al., 2016; Liu et al., 2017). Creating models that can work with stochastic environments is the motivation behind numerous recent papers: On one end of the complexity spectrum there are models like Video Pixel Networks (Kalchbrenner et al., 2017) and its variants (Reed et al., 2017b) that are powerful but computationally expensive models. These models generate a video frame-by-frame, and generate each frame pixel-by-pixel. SV2P (Babaeizadeh et al., 2018) and SAVP (Lee et al., 2018) do not model each individual pixel, but autoregressively generate a video frame-by-frame. On the other end of the spectrum, sSSM (Buesing et al., 2018) is a faster model that generates an abstract state at each time step which can be decoded into a frame when required. All these

stochastic models still generate frames (or states) one time-step at a time and the input and output frames must be consecutive. By bypassing these two issues JUMP extends this spectrum of models as a flexible and faster stochastic model. Additionally, unlike prior methods, our models can be used for a wider range of tasks – we demonstrate our approach on stochastic 3D reconstruction.

**3D Reconstruction** Research in 3D reconstruction can be broadly divided into two categories: on the one hand there are traditional 3D reconstruction methods that require some predefined description of the underlying 3D structure. This type of algorithms include structure-from-motion, structure-from-depth and multi view geometry techniques and while these models have achieved impressive results, they require researchers to handcraft a feature space in advance (Pollefeys et al., 2004; Wu et al., 2016; Zhang et al., 2015).

On the other hand there are more recent neural approaches that learn an implicit representation from data directly. The aim of these models however is usually to learn the distribution over images as opposed to learning the 3D structure of an environment (Kingma & Welling, 2014; Rezende et al., 2014; Goodfellow et al., 2014; Gregor et al., 2016b). Viewpoint transformation networks do focus on learning this structure, however the possible transformations are limited and the models have only been non-probabilistic so far (Choy et al., 2016; Tatarchenko et al., 2016; Fouhey et al., 2016).

Finally, GQN (Eslami et al., 2018) is a recent deep learning model used for spatial prediction. However, GQN was primarily used for deterministic 3D reconstruction. In stochastic 3D reconstruction, GQN is unable to capture several correlated frames of occluded regions of a scene.

**Meta-Learning** Finally, our task can be framed as a few-shot density estimation problem and is thus related to some ongoing research in the area of meta learning. Meta learning is often associated with few-shot classification tasks (Vinyals et al., 2016; Koch et al., 2015) but these algorithms can be extended to few-shot density estimation for image generation (Bartunov & Vetrov, 2018). Additional approaches include models with variational memory (Bornschein et al., 2017), attention (Reed et al., 2017a) and conditional latent variables (Rezende et al., 2016). Crucially, while these models can sample the estimated density at random they cannot query it at specific target points and their application is limited to visually less challenging datasets like omniglot. Concurrent work on neural processes (Garnelo et al., 2018), can query the density at specific target points, but on low-dimensional data-sets like 2D function regression and 2D contextual bandit problems.

## 3 METHODS

### 3.1 PROBLEM DESCRIPTION

We consider problems where we have a collection of "scenes". Scenes could be videos, 3D spatial scenes, or in general any key-indexed collection. A scene $S^{(i)}$ consists of a collection of viewpoint-frame (key-value) pairs $(v_1^{(i)}, f_1^{(i)}), ..., (v_n^{(i)}, f_n^{(i)})$ where $v_j^{(i)}$ refers to the indexing 'viewpoint' information and $f_j^{(i)}$ to the frame. For videos the 'viewpoints' are timestamps. For spatial scenes the 'viewpoints' are camera positions and headings. For notational simplicity, we assume that each scene has fixed length $n$ (but this is not a requirement of the model). The viewpoint-frame pairs in each scene are generated from a data generative process $D$, as formulated below. Note that the viewpoint-frame pairs are typically not independent.

$$(v_1^{(i)}, f_1^{(i)}), ..., (v_n^{(i)}, f_n^{(i)}) \sim D = P((v_1, f_1), ..., (v_n, f_n)) \tag{1}$$

Each scene $S$ is split into a context and a target. The context $C$ contains $m$ viewpoint-frame pairs $C = \{(v_i, f_i)\}_{i=1}^m$. The target $T$ contains the remaining $n - m$ viewpoints $V = \{v_i\}_{i=m+1}^n$ and corresponding target frames $F = \{f_i\}_{i=m+1}^n$. At evaluation time, the model receives the context $C$ and target viewpoints $V$ and should be able to sample possible values $\hat{F}$ corresponding to the viewpoints $V$. In particular, the model parameterizes a (possibly implicit) conditional distribution $P_\theta(F|C, V)$, from which the frames are sampled.

Given a training set of $N$ example scenes from data distribution $D$, the training objective is to find model parameters $\theta$ that maximize the log probability of the data

$$\mathop{\mathbb{E}}_{C,V,F \sim D}[\log P_\theta(F|C, V)]. \tag{2}$$

## 3.2 MODEL (GENERATION)

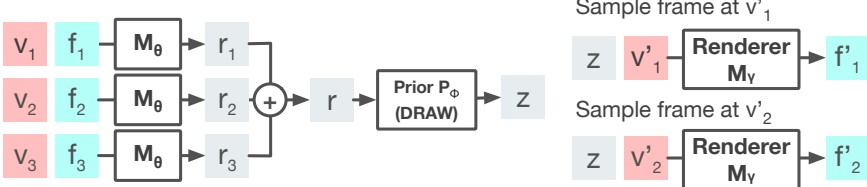

Figure 1: JUMP uses a prior $P_\phi$ to sample a latent $z$, conditioned on the input frames and viewpoints. JUMP then uses a rendering network $M_\gamma$ to render $z$ at arbitrary viewpoints e.g. $v'_1, v'_2$.

We implement JUMP as a latent variable model. For each scene $S$, JUMP encodes the $m$ viewpoint-frame pairs of the context $C$ by applying a representation function $M_\theta$ to each pair independently. The resulting representations $r_1, ..., r_m$ are aggregated in a permutation-invariant way to obtain a single representation $r$. The latent variable $z$ is then sampled from a prior $P_\phi$ that is conditioned on this aggregated representation $r$. The idea behind $z$ is that it can capture global dependencies across the $n - m$ target viewpoints, which is crucial to ensure that the output frames are generated from a single consistent plausible scene. For each corresponding target viewpoint $v_i$, the model applies a deterministic rendering network $M_\gamma$ to $z$ and $v_i$ to get an output frame $f_i$. Our model, JUMP, can thus be summarized as follows.

$$r_j = M_\theta(v_j, f_j) \text{ for all } j \in 1, ..., m$$
$$r = r_1 + ... + r_m$$
$$z \sim P_\phi(z|r)$$
$$f_i = M_\gamma(z, v_i) \text{ for all } i \in m + 1, ..., n$$

In JUMP, the representation network $M_\theta$ is implemented as a convolutional network. The latent $z$ is sampled using a convolutional DRAW (Gregor et al., 2015; 2016b) prior, an LSTM-like model that recurrently samples latents over several iterations. The rendering network $M_\gamma$ is implemented as an LSTM where the inputs $z$ and $v$ are fed in at each recurrent step. We give details of these implementations in the Appendix. Note that these building blocks are easily customizable. For example, DRAW can be replaced with a regular variational autoencoder, albeit at a cost to the visual fidelity of the generated samples.

## 3.3 MODEL (TRAINING)

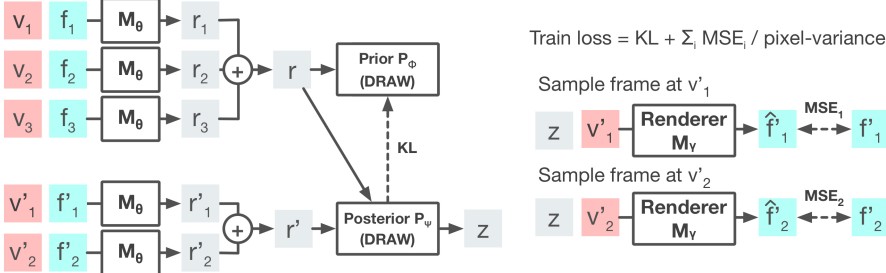

Figure 2: JUMP is trained using an approximate posterior $P_\psi$ that has access to multiple targets.

We wish to find parameters $\theta^*, \phi^*, \gamma^*$ that maximize the log probability of the data under our model:

$$\theta^*, \phi^*, \gamma^* = \arg\max_{\theta,\phi,\gamma} f(\theta, \phi, \gamma) = \arg\max_{\theta,\phi,\gamma} \mathbb{E}_{C,V,F \sim D} [\log P_{\theta,\phi,\gamma}(F|C,V)]. \quad (3)$$

Since this optimization problem is intractable we introduce an approximate posterior $P_{\psi,\theta}(z|C, V, F)$. We train the model by maximizing the evidence lower bound (ELBO), as

in (Rezende et al., 2014; Kingma & Welling, 2014), see derivation details in the Appendix. The resulting formulation for the ELBO of our model is as follows.

$$\mathbb{E}_{C,V,F\sim D}\left[\mathbb{E}_{z\sim P_{\psi,\theta}(z|C,V,F)}\left[\log\prod_{i=m+1}^{n}P_{\gamma}(f_i|z,v_i)\right]-\text{KL}(P_{\psi,\theta}(z|C,V,F)\,||\,P_{\phi,\theta}(z|C))\right]. \quad (4)$$

Note that this expression is composed of two terms: the reconstruction probability and the KL-divergence from the the approximate posterior $P_{\psi,\theta}(z|C,V,F)$ to the conditional prior $P_{\phi,\theta}(z|C)$. We use the reparameterization trick to propagate gradients through the reconstruction probability. As we are considering Gaussian probability distributions, we compute the KL in closed form. For training, to ensure that our model's likelihood has support everywhere, we add zero-mean, fixed variance Gaussian noise to the output frame of our model. This variance (referred to as the *pixel-variance*) is annealed down during the course of training.

## 4 EXPERIMENTS

We evaluate JUMP against a number of strong existing baselines on two tasks: a synthetic, combinatorial video prediction task and a 3D reconstruction task.

### 4.1 VIDEO PREDICTION

**Narratives Dataset:** We present quantitative and qualitative results on a set of synthetic datasets that we call "narrative" datasets. Each dataset is parameterized by a "narrative" which describes how a collection of shapes interact. For example, in the "Traveling Salesman" narrative (Figure 4), one shape sequentially moves to (and "visits") 4 other shapes. A dataset consists of many videos which represent different instances of a single narrative. In each instance of a narrative, each shape has a randomly selected color (out of 12 colors), size (out of 4 sizes), and shape (out of 4 shapes), and is randomly positioned. While these datasets are not realistic, they are a useful testbed. In our Traveling Salesman narrative with 5 shapes, the number of distinct instances (ignoring position) is over 260 billion. With random positioning of objects, the real number of instances is higher.

**Flexibility of JUMP:** JUMP is more flexible than existing video prediction models. JUMP can take arbitrary sets of frames as input, and directly predict arbitrary sets of output frames. We illustrate this in Figure 3. In this "Color Reaction" narrative, shape 1 moves to shape 2 over frames 1 - 6, shape 2 changes color and stays that color from frames 7 - 12. Figure 3A shows the ground truth narrative. JUMP can take two frames at the start of the video, and sample frames at the end of the video (as in Figures 3B, 3C). Alternatively, JUMP can go "backwards" and take two frames at the end of the video, and sample frames at the start of the video (as in Figures 3D, 3E).

**Quantitative Comparisons:** We quantitatively compare the image frame quality in JUMP with sSSM (Buesing et al., 2018), SV2P (Babaeizadeh et al., 2018), and CDNA (Finn et al., 2016). The other models cannot do jumpy prediction. So we compare these models when used for forward prediction on the Traveling Salesman narrative dataset, where one shape sequentially moves to (and "visits") 4 other shapes. The training set contains 98K examples, and the test set contains 1K examples. To evaluate each model, we take 30 sample continuations for each video and compute the minimum mean squared error between the samples and the original video. This metric, similar to the one in (Babaeizadeh et al., 2018), measures that the model can (in a reasonable number of samples) sample the true video. A model that exhibits mode collapse would fare poorly on this metric because it would often fail to produce a sample close to the original video.

We swept over model size, learning rate, and stochasticity parameters for each model (see the Appendix for more details). We selected the best hyperparameter configuration. We ran the model with that hyperparameter configuration over 15 random seeds (10 for CDNA), an advantage of testing on a simple synthetic dataset. We ran all models for 3 days using distributed ADAM on 4 Nvidia K80 GPUs. For sSSM, we discarded runs where the KL loss became too low or too high (these runs had very bad metric scores), and for SV2P we discarded a run which had especially poor metric scores. This discarding was done to the benefit of SV2P and sSSM – for JUMP we used all runs. The plot, with error bars of $\sqrt{2}$ times the standard error of the mean, is shown in Figure 5a.

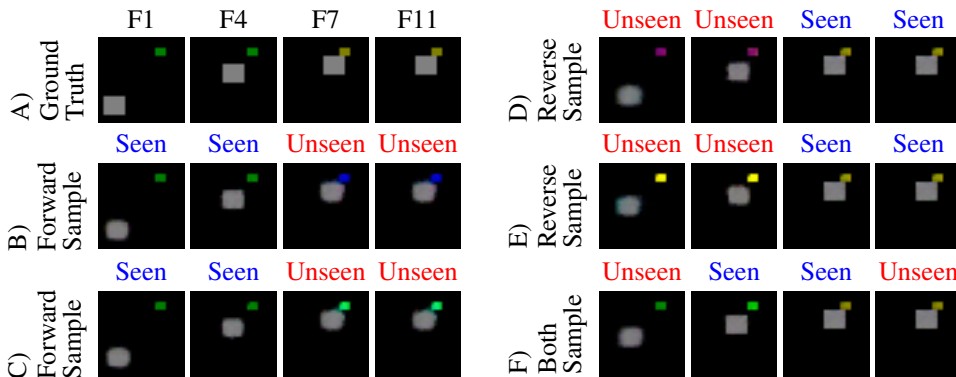

Figure 3: In the "Color Reaction" narrative, shape 1 moves to shape 2 over frames 1 - 6, shape 2 changes color and stays that color from frames 7 - 12. The ground truth is shown in sub-figure A. Our model sees 2 frames, labeled 'seen', and samples 2 'unseen' samples. Our model is flexible with respect to input-output structure, and can roll a video forwards (Figures 3B, 3C), backwards (Figures 3D, 3E), or both ways (Figure 3F).

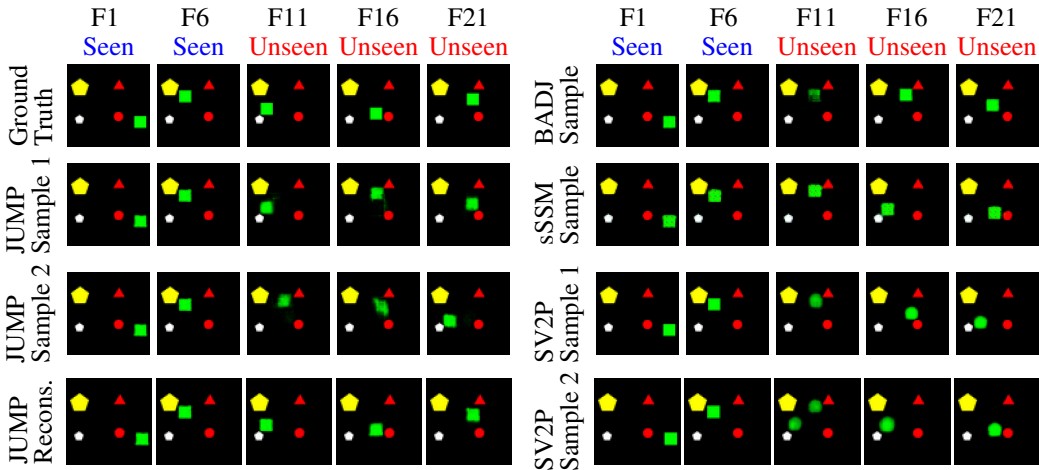

Figure 4: In the Traveling Salesman narrative, one shape (in this case the green square) sequentially moves towards and visits the other four shapes in some stochastic order. JUMP, sSSM and SV2P generally capture this, but with shape and color artifacts. BADJ is JUMP without the architectural features used to enforce consistency between target frames. BADJ does not capture a coherent narrative where the green square visits all four objects (in this case the white pentagon is not visited).

The plot shows that JUMP converges much more reliably. Averaged across runs, our model performs significantly better than sSSM, SV2P, and CDNA. This is despite discarding the worst runs for sSSM and SV2P, but keeping all runs of JUMP. One possible hypothesis for why our model converges more reliably is that we avoid back-propagation through time (BPTT) over the video frames. BPTT is known to be difficult to optimize over. Instead, JUMP tries to reconstruct a small number of randomly chosen target frames, from randomly chosen context frames.

**Qualitative Comparisons:** In Figure 4, we show samples from JUMP, sSSM, and SV2P. We also include samples from JUMP without the architectural features used to enforce consistency between the target frames, that is, the deterministic rendering network, global query-independent latent, and reconstructing multiple target-frames simultaneously. This is labeled as 'BADJ'. We note that these architectural components are essential for JUMP. Without these, each sampled frame is independent and so the frames do not form a coherent video.

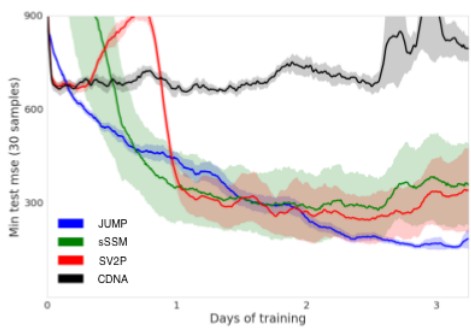 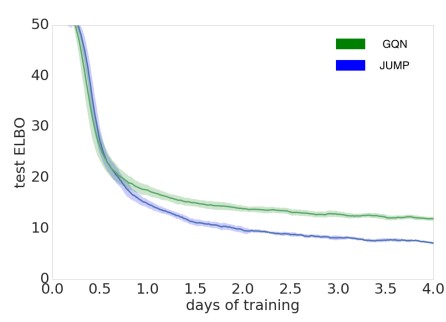

(a) JUMP converges more reliably to lower test errors than video prediction models, over 15 runs.

(b) JUMP achieves lower (better) negative test ELBO scores than GQN, over 6 runs.

Figure 5: Quantitative comparisons between JUMP and video and scene prediction models.

## 4.2 3D RECONSTRUCTION

Our model is also capable of consistent 3D reconstruction. In this setup, JUMP is provided with context frames $f_1, ..., f_m$ from a single 3D scene together with camera positions $v_1, ..., v_m$ from which those frames were rendered. The model is then asked to sample plausible frames rendered from a set of arbitrary camera positions $v_{m+1}, ..., v_n$. Often the model is asked to sample frames in an occluded region of the scene, where there are multiple possibilities for what the region might contain. Even in the presence of this uncertainty, JUMP is able to sample consistent frames that form a coherent scene. We encourage the reader to view videos visualizations of our experiments at https://bit.ly/2O4Pc4R.

**MNIST Dice Dataset:** To demonstrate this, we develop a 3D dataset where each scene consists of a cube in a room. Each face of the cube has a random MNIST digit (out of 100 digits) engraved on it. In each scene, the cube is randomly positioned and oriented, the color and textures of the walls are randomly selected, and the lighting source (which creates shadows) is randomly positioned. The context frames show at most three sides of the dice, but the model may be asked to sample camera snapshots that involve the unseen fourth side. We show quantitatively and qualitatively that JUMP performs better than GQN on this dataset, because GQN is unable to capture a coherent scene.

**JUMP vs GQN (Qualitative):** GQN samples each frame independently, and does not sample a coherent scene. We illustrate this in Figure 6, where we show an example scene from our test-set. The context frames (blue cones) see three sides of the cube, but the model is queried (red cones) to sample the occluded fourth side of the cube. Figure 6 also shows the samples for JUMP and GQN. GQN (right column) independently samples a 7 and then a 0 on the unseen side of the dice. JUMP samples a coherent scene, where the sampled unseen digit is consistently rendered across different viewpoints. JUMP's reconstruction accurately captures the ground truth digit, which shows that the model is capable of sampling the target.

**JUMP vs GQN (Quantitative):** We can compare GQN and JUMP by analyzing the test-set negative ELBO (as a proxy for the test-set negative log likelihood) over multiple target frames, each showing different viewpoints of the same unseen face of the cube. This serves as a quantitative measure for the quality of the models' reconstruction. To motivate why JUMP should do better, imagine that we have a perfectly trained GQN and JUMP, which captures all the nuances of the scene. Since there are 100 possible digits engraved on the unseen side, there is a 1/100 chance that each sampled frame captures the ground truth digit on the unseen face. GQN samples the unseen digit independently for each viewpoint, so the probability that a set of three frames all capture the ground truth digit is 1/1000000. On the other hand, JUMP captures the correlations between frames. If the digit is correct in one of three frames, it should be correct in the other two frames. So the probability that a set of three frames all capture the ground truth digit is 1/100. In other words, a perfectly trained consistent model will have better log likelihoods than a perfectly trained factored model.

In practice, the benefits of consistency may trade off with accuracy of rendering the scene. For example, JUMP could produce lower quality images. So it is important to compare the models' performance by comparing the test-set ELBOs. Figure 5b compares the test-set ELBOs for JUMP

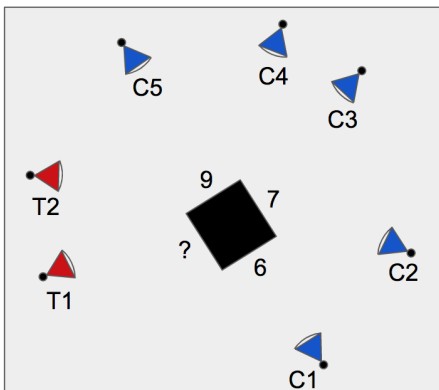 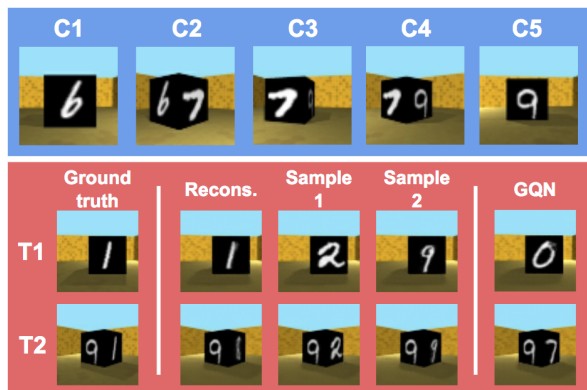

Figure 6: A cube in a room, with MNIST digits engraved on each face (test-set scene). The blue cones are where the context frames were captured from. The red cones are where the model is queried. The context frames see three sides of the cube, but the models are tasked to sample from the fourth, unseen, side. GQN (right column) independently samples a 0 and 7 for the unseen side, resulting in an inconsistent scene. JUMP samples a consistent digit (2 or 9) for the unseen cube face.

and GQN. We ran 7 runs for each model, and picked the best 6/7 runs for each model (1 run for JUMP failed to converge). The results suggest that JUMP achieve quantitatively better 3D reconstruction than GQN.

**JUMP Consistency Analysis:** We also analyze the consistency of JUMP. We measure the KL divergence from the posterior to the prior network in a trained JUMP model. We give JUMP a context comprising of frames that show three sides of the cube. We condition the posterior on one additional target frame that shows the unseen side of the cube, and compute the KL divergence from the posterior to the prior, $KL_1$. Alternatively, we condition the posterior on three additional target frames that show the unseen side of the dice, and compute the KL divergence from the posterior to the prior, $KL_3$. The 2 extra target frames added for $KL_3$ do not add any information, so $KL_3 \approx KL_1$ for a consistent model. On the other hand, for a factored model like GQN, $KL_3 = 3KL_1$. We trained 12 JUMP models, and the mean $KL_3$ was 4.25, the mean $KL_1$ was 4.19, and the standard deviation of $KL_3 - KL_1$ was 0.092. This suggests that JUMP's predictions are consistent.

## 5 CONCLUSION

We have presented an architecture for learning generative models in the visual domain that can be conditioned on arbitrary points in time or space. Our models can extrapolate forwards or backwards in time, without needing to generate intermediate frames. Moreover, given a small set of contextual frames they can be used to render 3D scenes from arbitrary camera positions. In both cases, they generate consistent sets of frames for a given context, even in the presence of stochasticity. One limitation of our method is that the stochastic latent representation is of a fixed size, which may limit its expressivity in more complicated applications – fixing this limitation and testing on more complex datasets are good avenues for future work. Among other applications, video prediction can be used to improve the performance of reinforcement learning agents on tasks that require looka-head (Racanière et al., 2017; Buesing et al., 2018). In this context, the ability to perform jumpy predictions that look many frames ahead in one go is an important step towards agents that can explore a search space of possible futures, as it effectively divides time into discrete periods. This is an avenue we will be exploring in future work.

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

# A  APPENDIX A: ELBO DERIVATION

The model is trained by maximizing an evidence lower bound, as in variational auto-encoders. We begin by expressing the log probability of the data in terms of the model's latent variable $z$.

$$\underset{C,V,F \sim D}{\mathbb{E}}[\log P(F|C,V)] = \underset{C,V,F \sim D}{\mathbb{E}}\left[\log \underset{z \sim P_{\phi,\theta}(z|C)}{\mathbb{E}}\left[P_\gamma(F|z,V)\right]\right]$$

The derivative of this objective is intractable because of the $\log$ outside the expectation. Using Jensen's inequality and substituting the $\log$ and expectation leads to an estimator that collapses to the mean of the distribution. Instead, we use the standard trick of parameterizing an approximate posterior distribution $P_{\psi,\theta}$. Instead of sampling $z$ from the prior $P_{\phi,\theta}(z|C)$ we sample $z$ from the approximate posterior $P_{\psi,\theta}(z|C,V,F)$ to get an equivalent objective.

$$f(\theta,\phi,\gamma) = f(\theta,\phi,\gamma,\psi) = \underset{C,V,F \sim D}{\mathbb{E}}\left[\log \underset{z \sim P_{\psi,\theta}(z|C,V,F)}{\mathbb{E}}\left[\frac{P_{\phi,\theta}(z|C)}{P_{\psi,\theta}(z|C,V,F)}P_\gamma(F|z,V)\right]\right]$$

Note that for efficiency, we typically sample a subset of the target viewpoint-frame pairs $(V,F)$ instead of conditioning the posterior and training on the entire set. We now apply Jensen's inequality to get a lower bound (ELBO) that we maximize as a surrogate.

$$g(\theta,\phi,\gamma,\psi) = \underset{C,V,F \sim D}{\mathbb{E}}\left[\underset{z \sim P_{\psi,\theta}(z|C,V,F)}{\mathbb{E}}\left[\log \left(\frac{P_{\phi,\theta}(z|C)}{P_{\psi,\theta}(z|C,V,F)}P_\gamma(F|z,V)\right)\right]\right]$$

$$g(\theta,\phi,\gamma,\psi) \leq f(\theta,\phi,\gamma,\psi)$$

We can split the ELBO into 2 terms, the reconstruction probability and the KL-divergence between the prior and posterior.

$$\text{RP}(\theta,\gamma,\psi) = \underset{C,V,F \sim D}{\mathbb{E}}\left[\underset{z \sim P_{\psi,\theta}(z|C,V,F)}{\mathbb{E}}\left[\log P_\gamma(F|z,V)\right]\right]$$

$$\text{KL}(\theta,\phi,\psi) = \underset{C,V,F \sim D}{\mathbb{E}}\left[\text{KL}(P_{\psi,\theta}(z|C,V,F) \,||\, P_{\phi,\theta}(z|C))\right]$$

$$g(\theta,\phi,\gamma,\psi) = \text{RP}(\theta,\gamma,\psi) - \text{KL}(\theta,\phi,\psi)$$

Since we consider Gaussian probability distributions, the KL can be computed in closed form. For the reconstruction probability, we note that each of the $n - m$ target frames $f_i$ are generated independently conditional on $z$ and the corresponding viewpoint $v_i$.

$$\text{RP}(\theta,\gamma,\psi) = \underset{C,V,F \sim D}{\mathbb{E}}\left[\underset{z \sim P_{\psi,\theta}(z|C,V,F)}{\mathbb{E}}\left[\log \prod_{i=m+1}^{n} P_\gamma(f_i|z,v_i)\right]\right]$$

We can then apply the standard reparameterization trick (where we sample from a unit Gaussian and scale the samples accordingly). This gives us a differentiable objective where we can compute derivatives via backpropagation and update the parameters with stochastic gradient descent.

# B  APPENDIX B: MODEL DETAILS AND HYPERPARAMETERS

We first explain some of the hyper-parameters in our model. For reproducibility, we then give the hyper-parameter values that we used for the narrative concepts task and the 3D scene reconstruction task.

For JUMP, recall that we added Gaussian noise to the output of the renderer to ensure that the likelihood has support everywhere. We call the variance of this Gaussian distribution the pixel-variance. When the pixel-variance is very high, the ELBO loss depends a lot more on the KL-divergence between the prior and the posterior, than the mean squared error between the target and

predicted images. That is, a small change $\delta$ in the KL term causes a much larger change in the ELBO than a small change $\delta$ in the mean squared error. As such, the training objective forces the posterior to match the prior, in order to keep the KL low. This makes the model predictions deterministic. On the other hand, when the pixel-variance is near zero, the ELBO loss depends a lot more on the mean squared error between the target and predicted images. In this case, the model allows the posterior to deviate far from the prior, in order to minimize the mean squared error. This leads to good reconstructions, but poor samples since the prior does not overlap well with the (possible) posteriors.

As such, we need to find a good pixel-variance that is neither too high, nor too low. In our case, we linearly anneal the pixel-variance from a value $\alpha$ to $\beta$ over 100,000 training steps. Note that the other models, sSSM and SV2P, have an equivalent hyper-parameter, where the KL divergence is multiplied by a value $\beta$. SV2P also performs an annealing-like strategy (Babaeizadeh et al., 2018).

For the traveling salesman dataset, we used the following parameters for the DRAW conditional prior/posterior net (Gregor et al., 2015; 2016a). The rendering network was identical, except we do not have a conditional posterior, making it deterministic.

| Name | Value | Description |
|---|---|---|
| nt | 4 | The number of DRAW steps in the network. |
| stride_to_hidden | $[2, 2]$ | The kernel and stride size of the conv. layer mapping the input image to the LSTM input. |
| nf_to_hidden | 64 | The number of channels in the LSTM layer. |
| nf_enc | 128 | The number of channels in the conv. layer mapping the input image to the LSTM input. |
| stride_to_obs | $[2, 2]$ | The kernel and stride size of the transposed conv. layer mapping the LSTM state to the canvas. |
| nf_to_obs | 128 | The number of channels in the hidden layer between LSTM states and the canvas |
| nf_dec | 64 | The number of channels of the conv. layer mapping the canvas state to the LSTM input. |
| nf_z | 3 | The number of channels in the stochastic latent in each DRAW step. |
| $\alpha$ | 2.0 | The initial pixel-variance. |
| $\beta$ | 0.5 | The final pixel-variance. |

For the encoder network, $M_\theta$, we apply a convolutional net to each image separately. The convolution net has 4 layers, with a ReLU non-linearity between each layer (but not after the last layer). The first layer has 8 channels, kernel shape of 2x2, and stride lengths of 2x2. The second layer has 16 channels, kernel shape of 2x2, and stride lengths of 2x2. The third layer has 32 channels, kernel shape of 3x3, and stride length of 1x1. The final layer has 32 channels, kernel shape of 3x3, and stride length of 1x1.

For all other datasets, we use the same encoder network, and similar hyper-parameters. For the MNIST Cube 3D reconstruction task, the main differences are that we use nt: 6, nf_to_hidden: 128, nf_dec: 128. We also had to use a slightly different annealing strategy for the pixel-variance. Simply annealing the variance down led to the KL-values collapsing to 0, and never rising back up. In other words, the predictions became deterministic. We use an annealing strategy somewhat similar to (Babaeizadeh et al., 2018). We keep the pixel-variance at 2.0 for the first 100,000 iterations, then keep it at 0.2 for 50,000 iterations, then keep it at 0.4 for 50,000 iterations, and then finally leave it at 0.9 until the end of training. The intuition is to keep the KL high first so that the model can make good deterministic predictions. Then, we reduce the pixel-variance to a low value (0.2) so that the model can capture the stochasticity in the dataset. Finally, we increase the pixel-variance so that the prior and the posteriors are reasonably similar.

Note that for each stochastic video prediction model we tested (CDNA, SV2P, sSSM), we swept over hyper-parameters, doing a grid search. We swept over size parameters, the learning rate, and the parameter used to control the KL divergence between the conditional posterior and the conditional prior. We ensured that we had tested hyper-parameter values slightly above, and below, the ones that we found worked best.

