# OpenReview forum: "Consistent Jumpy Predictions for Videos and Scenes"
_ICLR.cc/2019/Conference_

### Official Review · AnonReviewer1 · 2018-11-04
**Interesting approach, but not enough experimental evaluation**

**Rating:** 5
**Confidence:** 4

**Review:**

This paper presents a method for predicting future frames of a video (or unseen views of a 3D scene) in a "jumpy" way (you can query arbitrary viewpoints or timesteps) and "consistent" way (when you sample different views, the scene will be consistent). They use a VAE that encodes the input video in a permutation invariant way, which is achieved by summing the per-frame latent vectors. Then, they sample a latent vector using a DRAW prior. This latent vector can then be used to render the video/scene from different times/viewpoints via an LSTM decoder. They test the model on several toy datasets: they compare to video prediction methods on a dataset of moving shapes, and 3D viewpoint prediction on a 3D MNIST "dice" dataset.

Pros:
- The idea of developing new methods for viewpoint and video synthesis that allow for "jumpy" and "consistent" predictions is an important problem.

- The paper is fairly well written.

- The design of the model is reasonable (it is a natural extension of VAE viewpoint/future prediction methods).

Cons:
- All of the experiments were done on toy datasets. These are also not well-established toy datasets, and seem tailored to debugging the model, so it is not particularly surprising that the method worked. Since the main contribution is not very novel from a technical perspective (it is more about designing a model from existing, well-established components), this is a significant limitation.

- The paper suggests with experiments that GQN generates predictions that are less consistent across samples, but it is not clear exactly which design decisions lead to this difference. Why is this model more jumpy and consistent than GQN?

- The paper claims that JUMP trains more reliably than several video prediction methods in Figure 5. Yet, in the 3D viewpoint synthesis task, they suggest that JUMP had trouble with convergence, i.e.: "We ran 7 runs for each model, and picked the best 6/7 runs for each model (1 run for JUMP failed to converge)." This is confusing for two reasons. First, why was this evaluation protocol chosen (i.e. running 7 times and picking the 6 best)? If it was a post-hoc decision to remove one training run, then this should be clarified, and the experiment should be redesigned and rerun. Second, is the implication that JUMP is more stable than video prediction methods, but not necessarily more stable than GQN for viewpoint prediction?

- The paper should consider citing older representation learning work that deals with synthesizing images from multiple viewpoints. For example:

M. Tatarchenko, A. Dosovitskiy, T. Brox. "Multi-view 3D Models from Single Images with a Convolutional Network". ECCV 2016.

- There is insufficient explanation of the BADJ baseline. What architectural changes are different?

- The decision to use DRAW, instead of a normal VAE prior, is unusual and not explained in much detail. Why does this improve the visual fidelity of the samples?

Overall:
The paper does not present enough evidence that this model is better at jumpy/consistent predictions than other approaches. It is evaluated only on toy datasets: if the technical approach were more novel (and if it was clearer where the performance gains are coming from) then this could be OK, but it seems to be a fairly straightforward extension of existing models.

---

> ### Author Response · Authors · 2018-11-24
> **Consistency and where performance gains come from**
>
> Thank you for the review. We appreciate the positive comments on the importance of the problem, and the constructive feedback.
>
> We first address what we think is the most important point in your review: why the model is more consistent than GQN. Our model is designed in a principled way to produce consistent samples, while GQN fundamentally cannot produce consistent samples. The short answer is that GQN predicts each frame independently, while JUMP models correlations between predicted frames. We explained why this conceptually leads to better log-likelihood scores, so there is a principled reason for performance gains. We agree that the writing on this should have been clearer, and will edit that.
>
> If we ask GQN to predict multiple frames, each frame is independent conditioned on the inputs. So if GQN was asked to predict what is behind a wall/occlusion, it would sample different things each time. This is briefly described in a few parts of the paper, e.g. page 7 where we say “GQN samples the unseen digit independently for each (target) viewpoint”. So GQN simply cannot produce correlated samples in these situations.
>
> Our method fixes this problem in a principled way: 1. In our model we use a latent to model stochastic elements of the scene (for example what is behind an occlusion). The latent is shared among multiple predictions, and captures correlations between these predictions. 2. A deterministic renderer takes in the latent and a query and produces a corresponding target. The renderer should be deterministic, otherwise inconsistencies between predicted frames tend to get introduced at this stage. 3. At training time, the model is asked to predict multiple correlated targets with a shared latent. The variational posterior takes an encoding of multiple targets as well. This ensures that the shared latent actually learns to capture stochasticity in the scene, as opposed to stochasticity specific to a single predicted frame.
>
> The important point is that in our model, multiple frames are not predicted independently given the inputs. There is a shared latent that captures correlations between them. Theoretically, the approach of using a shared latent to model correlations is grounded by Di Finetti’s theorem. Since the viewpoint-frame sequence (during training) is exchangeable, Di Finetti’s theorem tells us that the observations are all conditionally independent with respect to some latent variable. The theorem does not tell us how to learn such a latent variable. Points (2) and (3) above are essential to learn this shared latent.
>
> On page 8, we give a detailed explanation for why consistency improves the log-likelihood scores as well. We believe this addresses the question of "if it was clearer where the performance gains are coming from". To conclude, we believe our method isn't simply extending an existing method. It gives a framework/design principles for constructing a consistent and jumpy model.

---

> ### Author Response · Authors · 2018-11-27
> **Other concerns**
>
> Thank you for your patience. We now respond to your other questions.
>
> - “All of the experiments were done on toy datasets…”
>
> We believe our novel contribution is our method of producing consistent samples. Traditionally, models achieve consistency by being autoregressive. We want a model that is both “jumpy” and “consistent”. We outlined the 3 key ingredients for consistency in our previous response, and we believe these are novel. These are fundamental ingredients that can be used with any architectural choices. (e.g. DRAW can be replaced with a VAE, conv nets with a different architecture, etc).
>
> Testing learned 3D scene reconstruction models on “realistic” data is rather difficult at present because of scalability issues. For example, the GQN paper only tests on synthetic "toy" datasets as well, many of similar complexity.
>
> In short, there is a principled (and novel) reason why our model should produce consistent samples. The 3D reconstruction experiments validate this, and are of a similar complexity to prior work.
>
> - “The paper claims that JUMP trains more reliably than several video prediction methods in Figure 5. Yet, in the 3D viewpoint synthesis task, they suggest that JUMP had trouble with convergence.”
>
> This is very subtle, and we believe we did not do a good job communicating this in the paper. We would like to distinguish between “trouble with convergence” and “achieving better test scores”.
>
> All these models (GQN, SV2P, sSSM, JUMP) sometimes diverge. Our (fairly standard) definition of “diverging” is that the KL drops to 0. The idea is that if the KL drops to 0, the model has failed to capture any stochasticity. This is easy to detect and fix in real life - we can check if the KL drops to 0, and if it does re-train the model.  As such, we dropped *all such failed runs where the KL drops to 0 for all models and never included them in plots*. This is analogous to re-starting training if the weights become infinity. We did document all such failed runs in the paper. All of the models suffer from this issue, so there is no way around it.*
>
> In the video prediction results we claim a different thing. We dropped all runs for SV2P and sSSM (the baselines) where they diverged, and dropped an outlier run where SV2P did very poorly. After dropping those runs, *but not dropping any runs for JUMP*, JUMP achieves better mean prediction scores, and this is statistically significant. So this could simply be summarized as: JUMP achieves better average predictions scores. In the paper we explain why this might be the case.
>
> We will fix the explanations. Does this address your concern?
>
> - “The decision to use DRAW, instead of a normal VAE prior, is unusual and not explained in much detail. Why does this improve the visual fidelity of the samples?”
>
> We use DRAW since it produces higher quality samples, as the results in the DRAW paper show. We believe DRAW is fairly widely used (the paper has 800 cites).
>
> We will also add the cite you mentioned on older representation learning work.
>
> If these responses address your main concerns, we hope you will consider revising your score. As you mention, we believe our paper formulates an important, original problem. We present a principled approach to achieving this, and we hope this spurs more work on this problem. We hope future work addresses the scalability limitations with all these models so they can be tested on more complex scene reconstruction datasets.
>
> Thanks,
> Authors
>
> Extra details about statistical testing:
> * We should have formalized the hypothesis we were testing. In our case, the null hypothesis is: the mean score for runs of JUMP that don't diverge is worse than or equal to the mean score for runs of GQN that don't diverge, where diverge is defined as when the KL goes below a *pre-defined constant* epsilon. This is a valid hypothesis to test, and the way to test it is to run each model a bunch of times, and restart training/drop the run if the KL goes below epsilon. We understand why "dropping runs" rings an alarm for the reviewer, as it should. If the definition of "diverge" depended on relative performance of the models, this would be an invalid statistical test. For example, if the test involved dropping a run with the "worst" KL, or dropping the run with the "worst" score, it would be invalid.
>
> If this are any doubts about the statistical methodology, please let us know. We don't want to go into too much technical/philosophical detail here, but we are happy to provide more detailed explanations, since this is a nuanced issue. For example, a lot of RL papers take the "best 6/12" runs when comparing 2 models. This is never acceptable, because there isn't a meaningful hypothesis being tested. If they wanted to focus on comparing the best 50% of runs, the right way to test this hypothesis is to *randomly* partition the 12 runs into 6 groups of 2, and drop 1 run in each class.

---

### Official Review · AnonReviewer2 · 2018-11-05
**A nice but non-convincing trial for indexed data modeling.**

**Rating:** 4
**Confidence:** 4

**Review:**

This paper proposes a general method for indexed data modeling by encoding index information together with observation into a neural network, and then decode the observation condition on the target index. I have several concerns regarding the way the paper using indices, and the experimental result.

The strategy this paper use for indexed data is to encode all data in a black-box, which can be inefficient since the order of temporal data or the geometric structure of spatial data is not handled in the model. These orders can be essential to make reasonable predictions, since they may encode causal relations among those observations, and certainly cannot be ignored. Another critical problem for this paper is that the relative time scale are not explicitly modeled in the context. My worry is that when putting all those informative data into a black-box may not be the most efficient way to use them.

On the other hand, experiments in this paper look quite artificial. Since sequential and spatial modeling have multiple real-life applications. It would be great if this method can be tested on more real dataset.

This paper does show some promise on sequence prediction task in a long range, especially when the moving trace is non-linear. A reasonable uncertainty level can be seen in the toy experiments. And the sample quality has some improvement over competitors. For example, JUMP does not suffer from those multi-mode issues. These experiments can be further strengthened with additional numerical results.

For now, this paper does not convince me about its method for modeling general indexed data, both in their modeling assumption, and their empirical results. In my opinion, there is still a long way to go for challenging tasks such as video prediction. This paper proposes an extreme way to use indices, but it is still far from mature.

---

> ### Author Response · Authors · 2018-11-10
> **Encoder and modeling assumptions**
>
> Thank you for taking the time to review our paper.
>
> The main concern in this review is the design of our architecture, in particular that we do not hand craft our model to explicitly handle the “order of temporal data” or “geometric structure of spatial data”. We first want to note that we do not ignore "order of temporal data or the geometric structure of spatial data", all this is provided to the model, which has to learn to assimilate the information to make predictions.
>
> First, we believe there is a misunderstanding of the main point of our paper. We describe a framework for consistent but jumpy predictions, and architectural details are not the main point. Existing models for scene prediction (e.g. Generative Query Networks) produce independent samples that do not form a coherent/consistent scene (See Figure 6, column labeled GQN). Our method has a few key features to fix this problem, 1. Our model generates a stochastic latent conditioned on the context. This latent models the stochastic elements of the scene. 2. A deterministic renderer takes in the latent and a query and produces a corresponding target. The renderer should be deterministic, otherwise inconsistencies between predicted frames could be introduced at this stage. 3. To enforce consistency, the variational posterior takes an encoding of multiple targets. Future work can and should experiment with architectural details that better account for temporal and spatial structure, and they can combine these with the methods we introduce to get consistent, jumpy predictions.
>
> Second, theoretically, there aren’t any restrictive “modeling assumptions” in our approach to consistent, jumpy prediction. The approach is grounded by Di Finetti’s theorem. Since the viewpoint-frame sequence is exchangeable, the observations are all conditionally independent with respect to some latent variable. The theorem does not tell us how to learn such a latent variable. Points (2) and (3) above are essential to learn this shared latent.
>
> Third, we believe that it is actually a strength of our model that we do not need to handcraft features or temporal/spatial structure, but that the model learns these. An advantage of our approach is that we do not have to tune it to every specific domain - the same method can be applied for spatial prediction, temporal prediction, image in-painting, or even prediction in space and time together. While currently such models (including in prior work), don't handle complex scenes, we don't see this as a big drawback. It's analogous to say 10 years ago when neural networks were far worse than hand crafted methods at video prediction, text to speech, machine translation, object tracking, etc.  As with those domains, more work needs to be done to scale neural architectures for spatial modeling to real scenes, and we believe that will come over a series of works over the next few years.
>
> Do these address your concerns?
>
> Reference:
> S. M. Ali Eslami, Danilo Jimenez Rezende, et al. Neural scene representation and rendering. In Science 2018.

---

> ### Author Response · Authors · 2018-11-27
> **Real dataset**
>
> Thank you for your patience. We respond to your final concern.
>
> We agree that demonstrating our method on real data would be interesting. We note, however, that prior learned approaches (e.g. GQN) was only shown to work on synthetic, toy datasets. Since we are proposing a new problem/approach, and our focus is not on scaling the model to more visually complex scenes, we think it's important to take intermediate steps and analyze the results. We hope future work addresses scalability and extends these models to work on real world data.
>
> We hope you will consider revising your score if these responses address some of your concerns.

---

### Official Review · AnonReviewer3 · 2018-11-16
**Novel problem, reasonable evaluation**

**Rating:** 7
**Confidence:** 2

**Review:**

The paper motivates and provides a model to generate video frames and reconstructions from non-sequential data by encoding time/camera position into the model training. The idea is to allow the model to interpolate, and more importantly, extrapolate from frames and learn the latent state for multiple frames together. The same techniques are also applicable to 3d-reconstruction.  JUMP is very closely related to GQN with the main difference being that the randomness in JUMP is learned better using a "global" prior. The evaluation is reasonable on multiple synthetic experiments including a 3d-scene reconstruction specially created to showcase the consistency capabilities in a stochastic generation. Paper is mostly clear but more priority should be given to the discussion around convergence and the latent state.

To me, the 3d-reconstruction use-case and experiments are more convincing than the video generation. Interpolation between frames seems like an easier problem when specifically trained for. On the other hand, video algorithms trained on sequential prediction should be able to go forward or backward in time. Moreover, jumpy prediction throws away information (the middle frames) that might lead to a better latent state. The experiments also show certain frames where there seems to be a superposition of two frames. In this aspect, sSSM is better despite having worse video quality.

For video experiments, prediction of more complex video, with far-away frame predictions would solidify the experiments. The narratives seem somewhat limited to show what kind of advantage non-sequential context gives you.

Reliable convergence (less variance of training progress) of the method seems to be the strongest argument in favor of the JUMP. It is also unclear whether having a global latent variable is why it happens. More discussion about this should probably be included considering that JUMPy prediction seems to be the cause of this.

Better evaluation of the latent state might have presented a better understanding of what the model is really doing with different samples. For example, what is the model causes some frames to look like superpositions??

---

> ### Author Response · Authors · 2018-11-27
> **Addressing main concern about solution novelty**
>
> Thank you for the review and the constructive feedback.
>
> We begin by addressing your main concern. We believe there are a number of differences between GQN and JUMP. The main novelty of our work is a way of doing consistent jumpy prediction. There are 3 key ingredients, none of which are present in GQN.
>
> 1. As you mention, our model uses a “global” latent to model stochastic elements of the scene (for example what the scene looks like behind an occlusion). The latent is shared among multiple predictions, and captures correlations between these predictions. 2. A deterministic renderer takes in the latent and a query and produces a corresponding target. The renderer should be deterministic, otherwise inconsistencies between predicted frames tend to get introduced at this stage. 3. At training time, the model is tasked to predict multiple correlated targets with a shared latent. The variational posterior takes an encoding of multiple targets as well. This ensures that the shared latent actually learns to capture stochasticity in the scene, as opposed to stochasticity specific to a single predicted frame.
>
> We believe this goes beyond learning the randomness better. As you mentioned, the “global” latent is certainly a key ingredient, because it allows the model to capture correlations between predicted frames which GQN cannot. There is also a principled reason for this choice. Since the viewpoint-frame sequence during training is exchangeable, Di Finetti’s theorem tells us that the observations are all conditionally independent with respect to some latent variable. The theorem does not tell us how to learn such a latent variable. Points (2) and (3) above are essential to learn this shared latent successfully.

---

> ### Author Response · Authors · 2018-11-27
> **Other concerns**
>
>
> Video prediction: For video prediction, one advantage of jumpy prediction is speed. Our model can predict the final frame 12x faster than SV2P in a 40 frame video.
>
> Non-sequential contexts: Jumpy interpolation, with non-sequential contexts could be used in model based planning. Given an image of a goal state, and the initial state, we could produce plausible intermediate states/subgoals that could aid an RL agent in planning its actions. This is just one example - in general, we believe jumpy predictions in both space and time is a very important problem. As humans, many of our cognitive abilities come from the ability to fill in the gaps in our knowledge e.g. understand what events may come after, before, or between events we have seen. This allows to understand and plan in the world. Of course, much more work needs to be done to realize these visions. However, we think our work sets up a novel, important problem that we hope spurs future research in this area.
>
> Training variance: In the paper, we mention that one hypothesis for why there is less variance in training progress is that jumpy models avoid BPTT (back propagation through time), which is known to be difficult to train well. However, this is just a hypothesis, more should be done to investigate this.
>
> Superposition of states: This is a common problem with VAE-based models. For example, SV2P exhibits similar issues. sSSM also does exhibit this issue, see sample 2 in Traveling Salesman Instance 4 in the supplementary URL containing videos of the models. Some very recent work (Rezende and Viola) works on mitigating this issue.
>
> We believe we introduce an important, novel problem, and present a principled, novel approach to make consistent jumpy predictions. If these address your concerns, we hope you will consider revising your rating, or letting us know if you have any follow up questions.
>
> References:
> Danilo J. Rezende and Fabio Viola. Taming VAEs. Arxiv October 2018.

---

### Author Response · Authors · 2018-12-02
**Reviewer Responses**

Dear R1 and R2,

Thank you for reviewing our paper. We believe there were some initial misunderstandings about our contributions, but hope we addressed them in our response. If we have addressed these we hope you consider re-evaluating your judgement, and if not giving us feedback on how we can improve our work.

- The key point in R1's review was that it is not clear why our model is better at consistent/jumpy prediction. We have addressed this in detail in our response to R1. SV2P, sSSM are not jumpy models, and GQN is a factored model that predicts each image conditionally independently on the context. One main contribution of ours is a principled way to do consistent jumpy prediction. We introduce 3 key ingredients for this, and explained why they are important. We believe our contribution here is novel and principled. We also believe that we formulate a novel and important problem.

- One issue raised was about our datasets. We are very transparent that currently neural scene reconstruction methods don’t scale well to complex scenes. Prior scene reconstruction work (e.g. GQN) was also evaluated on similar complexity (but deterministic) datasets. We agree that a lot more work needs to be done to scale this line of work to complex scene datasets. As with fields like machine translation, text to speech, etc, we believe this will happen over many papers over the next few years. Our work is on consistent jumpy prediction and is complementary to the scalability issue, so we believe it’s important to take stepping stones and test on 'toy' datasets. The alternative is to not do work of this sort until scalability is resolved.

Thank you,
Authors

---

### Meta-Review · Area_Chair1 · 2018-12-17
**interesting model, weak experimental section**

**Confidence:** 4
**Recommendation:** Reject

**Metareview:**

This paper proposes a probabilistic model for data indexed by an observed parameter (such as time in video frames, or camera locations in 3d scenes), which enables a global encoding of all available frames and is able to sample consistently at arbitrary indexes. Experiments are reported on several synthetic datasets.

Reviewers acknowledged the significance of the proposed model, noted that the paper is well-written, and the design choices are sounds. However, they also expressed concerns about the experimental setup, which only includes synthetic examples. Although the authors acknowledged during the response phase that this is indeed a current limitation, they argued it is not specific to their particular architecture, but to the task itself. Another concern raised by R1 is the lack of clarity in some experimental setups (for instance where only a subset of the best runs are used to compute error bars, and this subset appears to be of different size depending on the experiment, cf fig 5), and the fact that the datasets used in this paper to compare against GQNs are specifically designed.

Overall, this is a really borderline submission, with several strengths and weaknesses. After taking the reviewer discussion into account and making his/her own assessment, the AC recommends rejection at this time, but strongly encourages the authors to resubmit their work after improving their experimental setup, which will make the paper much stronger.